# Perception of Health and Its Predictors Among Saudis at Primary Healthcare Settings in Riyadh: Insights from a Cross-Sectional Survey

**DOI:** 10.3390/healthcare13050464

**Published:** 2025-02-21

**Authors:** Seema Mohammed Nasser, Mamdouh M. Shubair, Amani Alharthy, Badr F. Al-Khateeb, Nouf Bin Howaimel, Mohammed AlJumah, Khadijah Angawi, Lubna Alnaim, Noof Alwatban, Abdulrahman Fayssal Farahat, Ashraf El-Metwally

**Affiliations:** 1Department of Nursing, College of Nursing, King Saud bin Abdulaziz University for Health Sciences, Riyadh 11481, Saudi Arabia; nasserse@ksau-hs.edu.sa; 2King Abdullah International Medical Research Center, Riyadh 11481, Saudi Arabia; khateebb@ngha.med.sa (B.F.A.-K.); elmetwally.ashraf@outlook.com (A.E.-M.); 3Ministry of the National Guard—Health Affairs, Riyadh 11426, Saudi Arabia; 4School of Health Sciences, Faculty of Human and Health Sciences (FHHS), University of Northern British Columbia (UNBC), 3333 University Way, Prince George, BC V2N 4Z9, Canada; 5Department of Health Science, College of Health and Rehabilitation Sciences, Princess Nourah bint Abdulrahman University, Riyadh 11671, Saudi Arabia; Afalharthy@pnu.edu.sa; 6Department of Epidemiology and Biostatistics, College of Public Health and Health Informatics, King Saud bin Abdulaziz University for Health Sciences, Riyadh 11481, Saudi Arabia; watbann@ksau-hs.edu.sa; 7King Abdulaziz Medical City, Ministry of National Guard—Health Affairs, Riyadh 11426, Saudi Arabia; 8Riyadh Second Health Cluster, Riyadh 11622, Saudi Arabia; nbinhowaimel@moh.gov.sa; 9King Fahd Medical City (KFMC), Ministry of Health (MOH), Riyadh 12231, Saudi Arabia; jumahm@gmail.com; 10Department of Health Services and Hospital Administration, Faculty of Economics and Administration, King Abdulaziz University, Jeddah 22254, Saudi Arabia; kkangawi@kau.edu.sa; 11Clinical Nutrition Department, College of Applied Medical Sciences, King Saud bin Abdulaziz University for Health Sciences, Al Ahsa 36428, Saudi Arabia; nuaimlu@ksau-hs.edu.sa; 12College of Medicine, Alfaisal University, Riyadh 11533, Saudi Arabia; abffarahat@alfaisal.edu

**Keywords:** self-perception of health, predictors, Saudi Arabia cross-sectional study

## Abstract

**Background/Objectives**: Despite a link between self-perception of health and morbidity and mortality, data are scarce on factors that can predict one’s health perception, particularly in nations like Saudi Arabia. We conducted a needs assessment to evaluate health perception and identify sociodemographic, behavioural, and comorbidity-related factors influencing health perception among Saudi individuals. **Methods**: We conducted a cross-sectional survey utilizing an electronic questionnaire that was distributed to 14,239 people who visited primary healthcare centers in Riyadh, Saudi Arabia. We used multiple logistic regression to identify predictors of good health. Data was analyzed using SPSS software. **Results**: About one-third of the individuals (33.7%) perceived to have excellent health and 35.6% perceived to have very good health. Only 2.1% of the study participants perceived to have poor health. Compared to participants younger than 50 years, those aged 50–75 years were 10% less likely to perceive their health as good (AOR: 0.89, 95% CI: 0.82, 0.97). Males were 1.09 times more likely to perceive their health as good than females (AOR: 1.09, 95% CI: 1.01, 1.18). Smokers were 74% less likely than non-smokers to perceive their health as good (AOR: 0.26; 95% CI: 0.24, 0.29). Obese individuals were 20% less likely to perceive their health in good condition than non-obese individuals (AOR: 0.80; 95% CI: 0.65, 0.98) Individuals with heart disease were about 50% less likely to perceive their health as good condition than those without heart disease (AOR: 0.52; 95% CI: 0.40, 0.76). **Conclusions**: Despite the high frequency of risk factors, we discovered that Saudis perceive their health to be good on average. However, an independent association between older age, females, smoking, obesity, and heart disease with poor health calls for future epidemiological studies incorporating qualitative dimensions to explore why these individuals with specific risk profiles perceive their health as worse than others.

## 1. Introduction

Population health is commonly assessed through indicators, such as morality, morbidity, and healthcare utilization [1,2]. However, evaluating health status from an individual’s perspective is an emerging area of interest in public and medical health [3]. Some scientific scholars recommend that in epidemiological studies, self-rated health may be considered an appropriate proxy for more objective health measures [4,5]. Other researchers suggest that owing to intricate interactions among social, emotional, and physical domains of health, ignoring self-perception of health may introduce bias when exploring underlying disease. This metric is particularly valuable for tailoring healthcare interventions and understanding patient needs [6].

Self-perception of health reflects an individual’s assessment across social, physical, mental, and emotional dimensions, and is often difficult for external evaluators to interpret [7]. Self-rating of health encompasses multiple dimensions and is easier to collect than objective health data. Moreover, self-rated health reflects an individual’s judgment about their quality of life and satisfaction with their health. Perceptions about health and life may add value to additional decisions to change one’s behaviour or attitude towards diseases or lifestyles [8]. The literature also suggests the importance of assessing self-perception of health based on a strong association between subjective health and various diseases or death from different diseases [9]. Moreover, self-rated health has also been consistently reported as a strong predictor of mortality in previously published cohort studies and their subsequent reviews and meta-analyses [10,11,12]. Indeed, the power to predict mortality increases by self-perception of health as people are more aware of their health conditions than external observers or healthcare providers [13].

Despite extensive research on health perceptions, there remains a gap in understanding the specific predictors of health perceptions among Saudis at primary healthcare settings in Riyadh. Recent studies have highlighted various factors influencing health perceptions, such as psychological well-being, social determinants of health, and cultural influences [14,15]. For example, Leite et al. (2019) found a bidirectional relationship between psychological well-being and health perception, suggesting that mental health significantly impacts how individuals perceive their health [15]. Additionally, Cooper et al. (2024) examined how social bonds and perceptions of health influence behaviors addressing social determinants of health, emphasizing the role of community support and self-efficacy [16]. Additionally, Mahjoob et al. (2024) conducted a systematic review on health-related quality of life in neurodivergent children, identifying various predictors, including environmental, demographic, and individual characteristics [17]. Despite the link between self-perception of health with morbidity and mortality, there is limited availability of data on factors that may predict one’s health perception, especially in countries such as Saudi Arabia. Hence, there is limited research specifically focusing on the Saudi population and the unique cultural and social factors that may influence health perceptions in this context. To address this knowledge gap, we undertook a need assessment study to assess the overall health perception of Saudi individuals. We also evaluated sociodemographic, behavioural risk factors, and comorbidities associated with health perception among Saudi individuals. The findings of the current study would allow policymakers to understand the factors or determinants of self-health perception to develop targeted strategies to improve self-perception of health among the Saudi population.

## 2. Materials and Methods

### 2.1. Study Design, Study Duration, and Study Setting

This was a cross-sectional survey, conducted from March 2023 to July 2023. The study is part of a broader health system reform initiative undertaken in Saudi Arabia between 2021 and 2022, transitioning from a traditional health system to health clusters. The Riyadh region consists of three health centers with primary health care centers (PHCs) and hospitals, all managed by Central Health Services. This cross-sectional study was completed in 48 of 105 PHCs.

### 2.2. Participant Selection and Sampling

Participants were selected using a multi-stage cluster sampling method to ensure a representative sample of individuals utilizing PHCs across the Riyadh region. The selection process followed these steps: At the first level, Riyadh was divided into three health clusters, all managed under Central Health Services. Health Cluster 2, which serves approximately 3.7 million inhabitants and consists of 105 PHCs, was chosen for the study due to its diverse population and extensive healthcare network. From this cluster, 48 PHCs were randomly selected using a stratified random sampling approach, ensuring proportional representation from different geographic areas, including urban and suburban centers. Within each selected PHC, eligible participants were identified through systematic random sampling. A list of scheduled patients visiting the PHC during the study period was obtained, and every nth patient was invited to participate. The sampling interval (n) was determined based on the daily patient volume and the required sample size. This method ensured that the sample was representative of the general population receiving primary healthcare services in Riyadh while minimizing selection bias.

### 2.3. Eligibility Criteria and Sample Size

The survey targeted adults aged 18 and above who visited the selected primary healthcare centers, encompassing both Saudi and non-Saudi participants. Furthermore, regardless of home status or country, anybody accessing primary healthcare centers was eligible to participate in the study. However, this study excluded staff from primary healthcare clinics and healthcare practitioners, as well as visitors under the age of 18. Patients with cognitive impairments preventing survey comprehension or those who refused to provide consent were excluded. The data collectors invited visitors to the waiting room to complete an electronic survey. The survey was completed by 14,239 research participants. Thus, the sample size for this analysis was 14,239 research participants.

### 2.4. Study Questionnaire Development and Description

The questionnaire used in this study was developed by the Central Health Services Reform Management Team in collaboration with consultants representing all regions of Saudi Arabia. As part of the health system reform initiative, a standardized tool was designed to assess health perceptions, behaviors, and priorities across all health clusters in the country. The questionnaire included multiple sections, covering self-reported health status; health priorities and concerns; health behaviors (smoking, fast food consumption, physical activity, alcohol use); health perception (ranging from excellent to poor); sociodemographic information (age, education, employment, gender, marital status); medical history and comorbidities (heart disease, diabetes, obesity, hypertension, hypercholesterolemia); and insurance coverage.

### 2.5. Study Questionnaire Validation and Reliability

To ensure validity and reliability, the questionnaire underwent several evaluation steps. Content validity was assessed by a panel of 15 experts, including healthcare professionals and public health specialists, who reviewed the questionnaire for relevance and clarity. Items were modified or removed based on their feedback. Face validity was evaluated through a pilot study conducted with a random sample of 200 participants. Participants were asked to provide feedback on question clarity, difficulty, and comprehension. Additionally, trained data collectors read questions aloud to participants during interviews, further confirming face validity. Reliability was assessed through test–retest reliability, where 100 individuals from the pilot study responded to the questionnaire again via phone after adjustments were made based on their initial feedback. The test–retest reliability coefficient was 0.83, indicating high reliability. To ensure linguistic accuracy, the questionnaire was translated from English to Arabic and back translated to English.

### 2.6. Pilot Study and Justification for Hail City Selection

The pilot study was conducted in Hail rather than Riyadh because the Central Health Services Reform Management Team designated Hail as the testing site within its own health cluster. This decision was based on the rationale that Hail represents the general Saudi population in terms of demographic and health characteristics. The pilot study included a sample of 100 patients and 20 participants in focus groups, who assessed question clarity and difficulty. Their feedback led to modifications before implementation in the Riyadh region and other health clusters across Saudi Arabia.

### 2.7. Data Collection

The cross-sectional survey was undertaken using an electronic survey in the presence of the interviewer. Initially, a questionnaire was created, and all questions were exported to an iPad or Android tablet for use by data collectors at primary healthcare institutions in Riyadh, Saudi Arabia. Before requesting people to participate in this vision screening study, data collectors checked their eligibility, and only people over the age of 18 were included. Trained data collectors administered the survey using an interview-based approach via iPads, ensuring accuracy and efficiency while minimizing missing data. The survey was conducted in person at PHCs, with participants providing responses based on their health perceptions and behaviors. Participation in the survey was entirely voluntary. Data collectors collected information using questionnaires from those who had provided informed consent. Data collection included sociodemographic characteristics (e.g., age, gender, number of household members, marital status, education level, employment status, and health status) and behavioural factors (e.g., smoking, use of fast or junk food, alcohol, and physical activity or exercise), as well as comorbidities (e.g., hypertension, diabetes, obesity, and COPD).

### 2.8. Statistical Methods

We generated descriptive variables after examining the distribution of variables with histograms and P–P plots. For example, normally distributed continuous variables such as age were presented with their means and standard deviations. Later, the age variable was grouped to determine the distribution of participants across different age groups. We presented frequencies and proportions for categorical variables including education, employment status, marital status, health status, and insurance coverage. In addition, univariate analysis was carried out using logistic regression with a *p*-value of 0.25. All factors with a *p*-value of 0.25 or lower were considered acceptable for multivariable logistic regression because the outcome of interest was binary (i.e., health perception: Yes/No). We used multivariable logistic regression to identify characteristics that substantially predicted vision screening (*p*-value < 0.05). All analyses were carried out with SPSS (Statistical Package for Social Sciences) software version 26. 

## 3. Results

### 3.1. Sociodemographic Characteristics of Study Participants

The sociodemographic data of the Saudi Arabian study participants are displayed in Table 1. Of the research participants, 43.3% were men, and 48.8% were aged 50–75 years. Of those surveyed, 51.5% said they had gone to college or university, while 34.7% of research participants were single, and 51.4% of adults were employed. About a quarter (24.3%) of the study participants had insurance, and 27.7% of them were smokers. Despite 60.7% of participants reporting regular physical activity, 5.2% were classified as obese.

### 3.2. Perception of Health Among Saudi Individuals

Figure 1 illustrates the perception of health among Saudi individuals. Approximately one-third (33.7%) perceived their health as excellent, while 35.6% perceived it as very good. Only 2.1% of the study participants rated their health as poor, as shown in Figure 1.

#### Differences in Health Perception Across Specific Demographic Groups

Table 2 presents the distribution of self-reported current health status across various demographic categories. The table provides a comprehensive overview of how self-perceived health status varies across these key demographic characteristics. For age, the largest proportion of individuals across all health status categories fell within the 50 to 75-year age range, followed by those younger than 50 years, and then those 75 years and older. Gender distribution revealed a slightly higher representation of females compared to males across all health status levels. Marital status showed that married individuals made up a larger portion of the sample compared to single individuals within each health status category. Similarly, employed individuals were more numerous than unemployed individuals across all health status groups. Finally, regarding education, the “College/University” category represented the largest segment of the population within each health status level, followed by “Up to High School”, “Others”, and then “Primary” education.

### 3.3. Sociodemographic Predictors of Good Health Among Saudis at Primary Healthcare Settings in Riyadh

Table 3 shows the predictors of good health among Saudis in primary healthcare settings in Riyadh. After adjusting for other sociodemographic variables, we found that compared to younger participants of <50 years, those who were older (50 to 75 years) were 10% less likely to perceive good health (AOR: 0.89, 95% CI: 0.82, 0.97). Males had 1.09 times higher odds of perceiving their health as good compared to females (AOR: 1.09, 95% CI: 1.01, 1.18). However, the findings of the multivariable analysis revealed that there was no significant relationship between education, marital status, employment status, and insurance coverage with perception of health.

### 3.4. Behavioural Risk Factors and Co-Morbidities Associated with Perception of Good Health

Table 4 illustrates the behavioural risk factors and co-morbidities associated with perception of good health. The findings of the final multivariate analysis (model 3) revealed that after adjusting for age, gender, and other co-morbidities in the model, smokers were 74% less likely to perceive their health as good than non-smokers (AOR: 0.26; 95% CI: 0.24, 0.29). Obese individuals were 20% less likely than non-obese individuals to perceive their health as in good condition (AOR: 0.80; 95% CI: 0.65, 0.98) after adjusting for age, gender, and other co-morbidities in the model. Likewise, individuals with heart disease were about 50% less likely to perceive their health as being in good condition than those without heart disease (AOR: 0.52; 95% CI: 0.40, 0.76) after adjusting for age, gender, and other variables in the model. However, we did not find a significant association between physical activity, fast food consumption, diabetes, and hypertension with the perception of good health as shown in Table 4.

## 4. Discussion

We undertook this cross-sectional study to study to assess the overall health perception of Saudi individuals along with identifying the risk factors associated with self-perception of health. We found that about one-third of the Saudi individuals perceived themselves to have excellent health, and one-third perceived themselves to have very good health. A very low proportion of the study participants perceived themselves to have poor health. We found that males were more likely to perceive their health as good than females. Older age, smoking, obesity, and heart disease emerged as significant predictors of poorer health perception.

Our findings align with Lakeh et al. (2015), who reported that over 77% of the respondents rated their health as excellent or very good in Saudi Arabia [18]. In addition, the authors also found that older age, female gender, and diagnosis with chronic diseases such as hypertension, diabetes mellitus, hypercholesterolemia, and limited physical activity were predictors of poor health perception. Interestingly, Lakeh et al. (2015) found weaker associations between smoking, obesity, and poor health, highlighting potential contextual differences or methodological variations [18]. A national survey from Pakistan revealed an independent association between tobacco use and poor health [19]. The strong negative association between smoking and perceived health in the current study reinforces existing knowledge regarding the detrimental effects of tobacco use and emphasizes the critical need for effective smoking cessation interventions tailored to the Saudi Arabian population, taking into account cultural attitudes towards smoking.

We did not find a strong association between hypertension and hypercholesterolemia with perception of health in multivariable analysis. Our age and gender-adjusted analyses revealed similar findings as reported by Lakeh et al. (2015) [18]. However, we did find a strong association between other communicable diseases such as obesity and heart disease with poor health. Similarly, our findings on the relationship between heart diseases and poor health are consistent with other studies where authors have found a distinct association between non-communicable diseases and poor health [20,21]. One plausible explanation of these consistent findings may be that after being diagnosed with chronic conditions, it is likely that people may not have access to good or universal health care. Hence, they perceived their health as being poor. The observed association between obesity and lower perceived health highlights the importance of comprehensive weight management strategies and public health initiatives that address the complex factors contributing to excess weight in Saudi Arabia, including dietary habits and cultural norms around food. The substantial impact of heart disease on perceived health underscores the need for effective disease management, preventative measures, and lifestyle modifications to improve both physical health and overall well-being in this population. Furthermore, culturally sensitive approaches to heart disease prevention and management are crucial in Saudi Arabia.

Also, age appears to play a role, as individuals in the 50–75-year age group were less likely to perceive their health as good compared to their younger counterparts, suggesting a potential decline in perceived health with increasing age. This highlights the need for targeted health promotion efforts aimed at older adults in Saudi Arabia, focusing on maintaining physical and mental well-being as individuals age, while also considering culturally relevant factors that may influence health perceptions and behaviors within this demographic. Our findings related to older age and female gender being associated with poor health are consistent with the findings of other studies [18,22]. These findings collectively suggest that as the age of an individual increases, the likelihood of good health perception declines. These consistent findings are not surprising and can be explained by the fact that older age usually leads to increasing ailments, disability, and hence poor perception of health than younger age [23]. Younger individuals often report better health, likely due to fewer age-related ailments. These findings are supported by previously published studies in the literature suggesting that an increase in age is associated with a rise in the number of diseases, disability, and hospitalization with subsequent poor perception of health [24].

Similarly, our findings regarding female gender being a significant predictor of poor health are consistent with prior studies [22,25]. For instance, a study conducted in Portugal documented that women were more likely to report poor health than men [25]. Poor subjective health might reflect a range of underlying problems among females, and differences in the rates of morbidity and mortality between men and women. Moreover, there is a greater tendency among females to report health problems than males, or these gender-related differences in health perception could be partially attributed to differences in socioeconomic status between females and males [26]. While we did not explore the reasons for poor health perception among females in the current study, this finding underscores the need for further research to explore the underlying causes of gender disparities in health perception within Saudi Arabia. Although males were slightly more likely than females to rate their health as good, the small magnitude of this difference warrants further investigation into potential gender-specific health behaviors, access to care, and societal factors influencing self-perception of health within the Saudi Arabian context. Cultural norms surrounding gender roles and responsibilities may play a part in how men and women perceive and report their health.

A major strength of this study is its large representative sample, enabling generalizability to similar populations in Saudi Arabia and beyond. We used multistage random sampling to reduce the issue of self-selection in the study, and every Saudi who visited a primary healthcare center had an equal chance of participating in the research. Second, we gathered information on a variety of questions, allowing us to investigate a wide range of sociodemographic, behavioural, health-related, and comorbid characteristics that potentially predict health perception and self-rating of health among Saudi citizens. Furthermore, data acquired via a validated and trustworthy questionnaire may have minimized the possibility of measurement error in the variables. This large-scale study provides valuable insights into predictors of health perception, offering data to assist researchers and policymakers in developing frameworks to improve healthcare access for Saudi individuals. However, our study’s findings must be regarded with caution due to significant limitations. First, because this was a cross-sectional electronic survey, temporality and causal relationships between predictors and perception of health cannot be confirmed. Second, despite strong test–retest validity, social desirability bias may have influenced participants’ reporting of sensitive behaviors. We endeavored to address this issue by ensuring that study participants’ data would be kept confidential. In our study, while a significant proportion of the Saudi population is covered by health insurance, the reported coverage was lower than expected. One possible reason for this discrepancy is the reliance on self-reported data, which may introduce recall bias or misunderstanding of insurance status among participants. Some individuals, particularly those receiving government-funded healthcare, may not perceive themselves as having formal insurance coverage. Additionally, certain population groups, such as expatriates with irregular employment or individuals outside the workforce, may have limited or no access to private insurance, contributing to the observed lower coverage rates.

## 5. Conclusions

Through this cross-sectional study, we found that Saudi individuals, overall, perceive their health as good despite the high prevalence of risk factors. The independent association of older age, female gender, smoking, obesity, and heart disease with poor health warrants future epidemiological studies incorporating qualitative dimensions. In addition, one should not ignore females and a strong association between female gender and poor health perception may likely indicate a myriad of health problems that need to be unpacked in future large studies. While many individuals perceive their health positively, the significant burden of risk factors necessitates targeted public health strategies to mitigate disparities in Saudi Arabia.

## 6. Practical Applications and Public Health Recommendations

The findings of this study offer several practical applications for public health initiatives in Saudi Arabia. Given the observed association between increasing age (50–75 years) and a decline in perceived health, targeted programs promoting healthy aging are crucial. These programs should emphasize regular physical activity, nutritional guidance, and accessible preventative health screenings, tailored to the specific needs and cultural context of older adults in Saudi Arabia. Furthermore, considering the strong negative impact of smoking and obesity on perceived health, intensified public health campaigns and accessible cessation programs are vital. These initiatives should be culturally sensitive, addressing social norms and beliefs surrounding tobacco use and dietary habits within Saudi Arabia. For individuals with heart disease, accessible and culturally appropriate disease management programs, including medication adherence support, lifestyle counseling, and cardiac rehabilitation, are essential to improve both physical health and perceived well-being. Finally, recognizing the influence of education on health perceptions, public health interventions should leverage community-based approaches to disseminate health information and promote health literacy, particularly among less formally educated populations. These initiatives should consider cultural values and beliefs to ensure effective communication and engagement. By translating these findings into actionable public health programs, we can work towards improving both the actual and perceived health of individuals in Saudi Arabia.

## 7. Key Findings

Our findings indicate a generally positive self-perception of health among the study participants, with a substantial proportion rating their health as excellent or very good. However, the notable segment of the population reporting less favorable health perceptions underscores the importance of understanding the contributing factors. Age appears to play a role, as individuals in the 50–75 years age group were less likely to perceive their health as good compared to their younger counterparts, suggesting a potential decline in perceived health with increasing age. This highlights the need for targeted health promotion efforts aimed at older adults, focusing on maintaining physical and mental well-being as individuals age. While males were slightly more likely to rate their health as good than females, the small magnitude of this difference warrants further investigation into potential gender-specific health behaviors, access to care, and societal factors influencing self-perception of health. The strong negative association between smoking and perceived health reinforces existing knowledge regarding the detrimental effects of tobacco use and emphasizes the critical need for effective smoking cessation interventions. Similarly, the observed association between obesity and lower perceived health highlights the importance of comprehensive weight management strategies and public health initiatives that address the complex factors contributing to excess weight. The substantial impact of heart disease on perceived health underscores the need for effective disease management, preventative measures, and lifestyle modifications to improve both physical health and overall well-being in this population. Further research exploring the interplay of these factors, alongside socioeconomic determinants of health, is crucial for developing targeted interventions and promoting positive health perceptions across all segments of the population.

## Figures and Tables

**Figure 1 healthcare-13-00464-f001:**
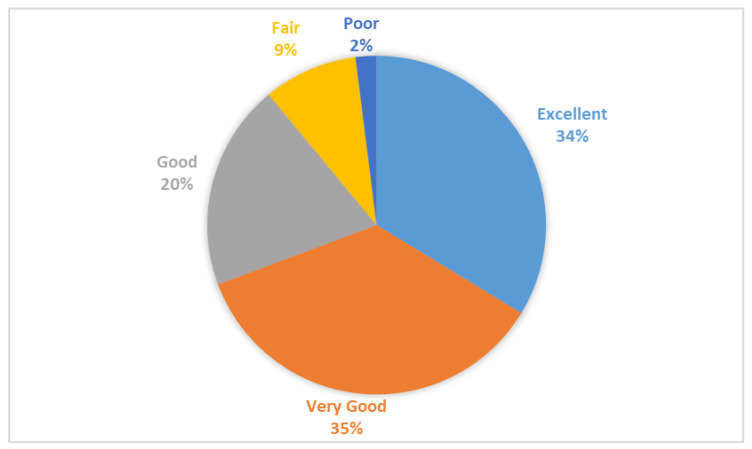
Perception of health among Saudi individuals.

**Table 1 healthcare-13-00464-t001:** Sociodemographic characteristics of study participants (n = 14,239).

	Frequency	Percentage
Age		
<50 years	4848	34.0
50 to 75 years	6945	48.8
At least 75 years	2446	17.2
Education		
Primary	572	4.0
Up to High School	3937	27.6
College/University	7336	51.5
Others	2394	16.8
Gender		
Female	8062	56.6
Male	6177	43.4
Marital status		
Not married	4939	34.7
Married	9300	65.3
Employment status		
Employed	7317	51.4
Unemployed	6922	48.6
Health status		
Excellent	4798	33.7
Very good	5076	35.6
Good	2815	19.8
Fair	1256	8.8
Poor	294	2.1
Insurance coverage		
Yes	3457	24.3
No	10,782	75.7
Smoking		
No	10,297	72.3
Yes	3942	27.7
Physical activity		
No	5598	39.3
Yes	8641	60.7
Obesity		
No	13,502	94.8
Yes	737	5.2

**Table 2 healthcare-13-00464-t002:** Differences in health perception across specific demographic groups.

Variable	Health Perception Based on Current Health Status					
	Category	Excellent	Very Good	Good	Fair	Poor
Age	<50 years	1915 (39.9%)	1610 (31.7%)	851 (30.2%)	367 (29.2%)	105 (35.7%)
	50 to 75 years	2312 (48.2%)	2481 (48.9%)	1408 (50.0%)	611 (48.6%)	133 (45.2%)
	At least 75 years	571 (11.9%)	985 (19.4%)	556 (19.8%)	278 (22.1%)	56 (19.0%)
Gender	Female	2441 (50.9%)	2875 (56.6%)	1743 (61.9%)	826 (65.8%)	177 (60.2%)
	Male	2357 (49.1%)	2201 (43.4%)	1072 (38.1%)	430 (34.2%)	117 (39.8%)
Marital Status	Married	2735 (57.0%)	3427 (67.5%)	2019 (71.7%)	909 (72.4%)	210 (71.4%)
	Single	2063 (43.0%)	1649 (32.5%)	796 (28.3%)	347 (27.6%)	84 (28.6%)
Employment Status	Employed	2564 (53.4%)	2519 (49.6%)	1469 (52.2%)	640 (51.0%)	125 (42.5%)
	Unemployed	2234 (46.6%)	2557 (50.4%)	1346 (47.8%)	616 (49.0%)	169 (57.5%)
Education	Primary	145 (3.0%)	180 (3.5%)	145 (5.2%)	76 (6.1%)	26 (8.8%)
	Up to High School	1427 (29.7%)	1289 (25.4%)	858 (30.5%)	294 (23.4%)	69 (23.5%)
	College/University	2547 (53.1%)	2561 (50.5%)	1354 (48.1%)	696 (55.4%)	178 (60.5%)
	Others	679 (14.2%)	1046 (20.6%)	458 (16.3%)	190 (15.1%)	21 (7.1%)

**Table 3 healthcare-13-00464-t003:** Sociodemographic predictors of health perception among Saudis at primary healthcare settings in Riyadh (n = 14,239).

Predictors	OR	95% CI		*p*-Value	AOR	95% CI		*p*-Value
		LL	UL			LL	UL	
Age								
<50 years	1			<0.001	1			0.005
50 to 75 years	0.99	0.91	1.07		0.89	0.82	0.97	
At least 75 years	1.19	1.08	1.32		1.05	0.93	1.18	
Education								
Primary	1			0.052	1			0.119
Up to High School	1.14	0.94	1.38		1.15	0.95	1.39	
College/University	1.03	0.85	1.24		1.06	0.88	1.28	
Gender								
Female	1			0.012	1			0.027
Male	1.10	1.02	1.18		1.09	1.01	1.18	
Marital status								
Single	1			0.375	NA			
Married	1.03	0.96	1.11					
Employment status								
Employed	1			0.643	NA			
Unemployed	1.02	0.95	1.09					
Insurance coverage								
No	1			0.652	NA			
Yes	0.98	0.90	1.07					

**Table 4 healthcare-13-00464-t004:** Behavioural risk factors and co-morbidities associated with health perception (n = 14,239).

Predictors	Model 1				Model 2				Model 3				
	OR	95% CI		*p*-Value	AOR	95% CI		*p*-Value	AOR	95% CI		*p*-Value	
		LL	UL			LL	UL			LL	UL		
Smoking													
No	1			<0.001	1			<0.001				<0.001	
Yes	0.26	0.23	0.28		0.25	0.23	0.28		0.26	0.24	0.29		
Physical activity													
No	1			<0.001	1			<0.001	1			0.88	
Yes	0.70	0.65	0.75		0.69	0.64	0.75		1.01	0.93	1.08		
Fast food consumption													
No	1			0.174	1			0.109	NA				
Yes	1.05	0.97	1.14		1.06	0.98	1.15						
Obesity													
No	1			<0.001	1			<0.001	1			0.031	
Yes	0.52	0.43	0.62		0.52	0.43	0.63		0.80	0.65	0.98		
Diabetes													
No	1			0.639	NA								
Yes	0.97	0.87	1.08										
Hypertension													
No	1			0.225	1			0.047	1			0.523	
Yes	0.93	0.83	1.04		0.89	0.79	0.99		1.05	0.91	1.2		
Hypercholesterolemia													
No	1			<0.001	1			<0.001	1			0.901	
Yes	0.76	0.67	0.86		0.74	0.65	0.84		0.99	0.85	1.15		
Heart disease													
No	1			<0.001	1			<0.001	1			<0.001	
Yes	0.34	0.27	0.43		0.34	0.27	0.43		0.52	0.40	0.76		

Model 1: Univariable analysis with unadjusted results. Model 2: Adjusted for age and gender. Model 3: Adjusted for age, gender, and other co-morbidities. OR: Odds ratio; AOR: Adjusted Odds ratio; CI: confidence interval; NA: Not applicable.

## Data Availability

Data is contained within the article.

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
