# Peer review of "Perception of Health and Its Predictors Among Saudis at Primary Healthcare Settings in Riyadh: Insights from a Cross-Sectional Survey"

_healthcare, 2025, doi:10.3390/healthcare13050464_

Round 1
Reviewer 1 Report
Comments and Suggestions for Authors
Dear Editor/Authors,
Thank you for the opportunity to review this manuscript. Here are my comments for your response.
Abstract
Introduction
It needs to be extended with updated references to reflect the gap in the literature.
Specific literature is needed to show what is found related to the predictors of health.
Methods
How were the participants selected randomly? More details are needed.
Study questionnaire validation and reliability
Where is the description of the questionnaire? What is its contents? Who developed it?
Did you do the validity of the questionnaire with the experts or the patients?
Why did you do the pilot study from Hail City not from Riyadh?
Page 3, line 133, it is appeared for me that the data collection were conducted as the handed questionnaire, is it?
Results:
Table 1, there is a wrong in the percentages.
Insurance Coverage 75.7%, I think most of people who live in Saudi Arabia are covered by health insurance. Can you explain this percentage in your study?
Lines 160 to 170 are unclear
Table 2, you need to consult a Statistician please, why there are two columns of 95 %CI? Where are the p-values?
Discussion
Should be updated with more discussion with updated studies.
References
Should be updated to the last 5 years
Best regards
Comments on the Quality of English LanguageAs above
Author Response
Dear Reviewer,
Thank you for all the suggestions; we have addressed all the comments in the manuscript. Please find our point-to-point response in the attached file.
Regards,

Reviewer 2 Report
Comments and Suggestions for Authors
Review in attachment.

Author Response

(The authors gave the same response as above.)

Round 2
Reviewer 1 Report
Comments and Suggestions for Authors
Thank you for your correspondence. No other comments.
Comments on the Quality of English LanguageGood